# A Comparative Study of Climatology, Energy and Mass Exchange in Two Forests on Contrasting Habitats in Central Siberia: Permafrost *Larix gmelinii* vs. Permafrost-Free *Pinus sylvestris*

Nadezhda M. Tchebakova [1,*], Viacheslav I. Zyryanov [1], Olga A. Zyryanova [1], Elena I. Parfenova [1], Takuya Kajimoto [2] and Yojiro Matsuura [3]

1 V.N. Sukachev Institute of Forest, Siberian Branch, Russian Academy of Sciences, 660036 Krasnoyarsk, Russia
2 Sado Island Center for Ecological Sustainability, Niigata University, Niigata 952-2206, Japan
3 Global Strategy Division, Forestry and Forest Products Research Institute, Tsukuba 305-8687, Japan
* Correspondence: ncheby@ksc.krasn.ru

**Abstract:** Inter-annual and seasonal variations of energy, vapor water, and carbon fluxes and associated climate variables in a middle taiga pine (*Pinus sylvestris*) forest on sandy soils and in a northern taiga larch (*Larix gmelinii*) forest on permafrost in central Siberia were studied from eddy covariance measurements acquired during the growing seasons of 1998–2000 and 2004–2008, respectively. Both the pure Scots pine of 215-year-old and pure Gmelin larch of 105-year-old forests naturally regenerated after forest fires, differed by their tree stand characteristics, and grew in extremely contrasting environments with distinctive climatic and soil conditions. Net radiation was greater in the pine forest due to higher values in the summer months and a longer growing season. Sensible heat flux was the larger term in the radiation balance in both forests. The Bowen ratio stayed between 1 and 2 during the growing season and was as high as 8–10 in dry spring in both forests. In the dry summers, latent heat explained 70%–80% of the daily net ecosystem $CO_2$ exchange (NEE) variation in both forests. The average NEE was significantly smaller in the larch ecosystem at $-4$ μmol m$^{-2}$s$^{-1}$ compared to $-7$ μmol m$^{-2}$s$^{-1}$ in the pine forest. NEP for the growing season was 83 in the larch forest on continuous permafrost and 228 g C m$^{-2}$ in the pine forest on warm sandy soils. Water use efficiency was 5.8 mg $CO_2$ g$^{-1}$H$_2$O in the larch forest and 11 mg $CO_2$ g$^{-1}$H$_2$O in the pine forest and appeared to be consistent with that in boreal forests. As a result of the forest structure change from Gmelin larch to Scots pine due to the permafrost retreat in a warming climate, the boreal forest C-sink may be expected to increase. Thus, potential feedback to the climate system in these "hot spots" of forest-forming replacement species may promote C-uptake from the atmosphere. However, as many studies suggest, in the pace of transition from permafrost to non-permafrost, C-sink would turn into C-source in hot spots of permafrost retreat.

**Keywords:** energy; water vapor and carbon flux; water use efficiency; energy; heat and water balance; permafrost; Scots pine; Gmelin larch; interior Siberia

## 1. Introduction

Central Siberia is located in the middle of the Eurasian continent, a massive land with a severe continental climate. The westerlies that travel a long distance from the Atlantic deliver little precipitation to Central Siberia. Additionally, permafrost, covering about 80% of Siberia, is a powerful factor controlling the forest and tree species distribution in the dry and cold climate in interior Siberia [1].

Permafrost limits the northward and eastward spread of Scots pine (*Pinus sylvestris* L.). Within the permafrost zone, pine can reach high latitudes on sandy soils along big river valleys where the permafrost may thaw deep down to 1.5–2 m [1–3]. Forests of Dahurian

larch (*Larix gmelinii* (Rupr.) Rupr. and *Larix cajanderi* Mayr) are related to permafrost and are found in central Siberia on the shallow soils which thaw as little as 30–40 cm during the growing season [4]. For more details on forest cover, see Supplementary File S1.

Scots pine and larch (*Larix* spp.) forests dominate the boreal coniferous forest in Siberia: they comprise 13% and 49% of its territory, respectively [1] and accumulate about 14,462 Tg C in their total phytomass, of which 25% is below ground [5]. Despite their abundance, relatively little is known about what a comparative role both Scots pine and permafrost larch forests play in biophysical (energy, water) and biogeochemical ($CO_2$) cycles at the subcontinental scale over Siberia. Analyses of carbon, water vapor, and energy fluxes at the ecosystem level have been conducted using the eddy-covariance method in a variety of Siberian boreal forest types: evergreen (Scots pine and dark conifers *Pinus sibirica, Abies sibirica*), small-leaved (birch) and mixed (birch-evergreen conifers) forests beyond the permafrost zone and deciduous conifer Dahurian (Gmelin + Cajander) larch forests within the permafrost zone. In the 1990s, the first eddy-covariance studies across Siberia were short-term, lasting from a month to one growing season [6–9]. Later, a few continuous eddy-covariance studies were conducted covering several years in both Scots pine (see Tellus Special issue 54B 2002) and Dahurian larch forests within the permafrost zone [10–16].

Among the most recent publications on Siberia are the following: Olchev et al. [17] which studied only net radiation, latent heat, sensible heat and net ecosystem exchange; Masyagina et al. [18] studied soil respiration; and [19,20] studied transpiration and water use efficiency in the same Gmelin larch and Scots pine forests. However, none of them directly and systematically compared the functional permafrost vs. non-permafrost parameters for these two contrasting forest habitats. We compared them and found that these ecosystem parameters are very similar after a decade or so. From the summaries of the global boreal forests of Virkkala et al. [21,22], we are aware of three forest eddy tower sites across Siberia: one non-permafrost site Zotino (in our study), two permafrost sites Tura (in our study) and Yakutia. A permafrost larch forest in Yakutia (site Spasskaya Pad) was discussed in multiple publications [10–12]. None of those publications compared permafrost forest ecosystems to non-permafrost ecosystems in detail. In our present study, we suggested the analyses of multiple ecosystem functional parameters (heat and water vapor fluxes: LE, H, β); $CO_2$ exchange (NEE, $R_{eco}$, and GPP); and water use efficiency (WUE) of two principal forest ecosystems: Scots pine and Gmelin larch forests in interior Siberia.

Recently, Arctic-boreal landscapes, including both permafrost and non-permafrost ecosystems, have been intensively studied based on eddy covariance measurements collected from many eddy sites across the circumboreal zone [23–27]. As predicted from general circulation models, profound warming is expected at high latitudes; thus, Arctic-boreal ecosystems become especially vulnerable to warming, which may transform the Arctic-boreal ecosystems from carbon sink to source [27]. Abbot et al. [28] analyzed 98 permafrost-region experts' evaluations of biomass, wildfire, and hydrologic carbon flux responses to climate change. Experts assessed that the permafrost region would become a carbon source to the atmosphere by 2100.

From 1990 to 2015, Virkkala et al. [21] compiled eddy covariance and chamber measurements of annual and growing season $CO_2$ fluxes of gross primary productivity (GPP), ecosystem respiration (ER), and net ecosystem exchange (NEE) from 148 terrestrial high-latitude sites to analyze the spatial patterns and drivers of $CO_2$ fluxes and test the accuracy and uncertainty of different statistical models. Virkkala et al. [21,22] developed a standardized monthly database of Arctic–boreal $CO_2$ fluxes (ABCflux) that aggregates in situ measurements of terrestrial net ecosystem $CO_2$ exchange and its derived partitioned component fluxes: gross primary productivity and ecosystem respiration. ABCflux can be used in a wide array of empirical, remote sensing and modeling studies to improve understanding of the regional and temporal variability in $CO_2$ fluxes and to better estimate the terrestrial ABZ $CO_2$ budget. ABCflux is openly and freely available online [21].

Here, we hypothesized that permafrost is a powerful driver that changes site climatology, vegetation cover, and energy and mass fluxes between vegetation and the atmosphere and thus may change the contemporary C-sink of the boreal forest into C-source if methane emissions exceed the C-sink with permafrost retreat, as many studies suggest [28].

Our goals were to (1) compare inter-annual and seasonal variations of energy and mass (water vapor and $CO_2$) fluxes evaluated from eddy-covariance measurements and associated climate variables and to infer the differences in the $CO_2$-exchange driven by various environmental factors in two central Siberian forests located in contrasting habitats: a *Larix gmelinii* forest on permafrost representative of the vast permafrost zone over tablelands in central Siberia and a *Pinus sylvestris* forest on comparatively warm sandy soils representative of adjacent West Siberia Plain. (2) try to expertly assess a central Siberian boreal forest C-sink change and potential feedback to the climate system in "hot spots" of the replacement of Gmelin larch forests on permafrost by the Scots pine forest advance followed by the permafrost retreat in a changing climate which would require additional research in interior Siberia.

## 2. Material and Methods

### 2.1. Site Description

Our experimental forest stands with eddy towers were established in: a pure larch (*Larix gmelinii*) forest representative of the northern taiga region of central Siberia and a pure pine forest (*Pinus sylvestris*, hereafter as pine) representative of the middle taiga (Figure 1). Both forest ecosystems were at a post-fire successional stage and were very distinctive in their habitat and tree stand characteristics.

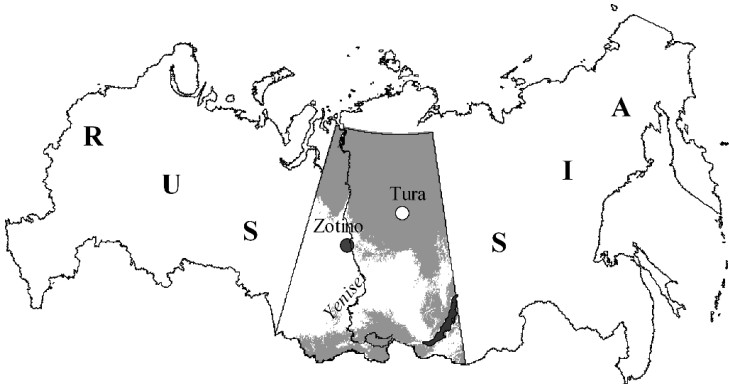

**Figure 1.** Two study sites, Zotino and Tura, in central Siberia. The continuous permafrost area is shaded.

The larch forest was located near the settlement of Tura, 600 km east of the Yenisei River, in the center of the Central Siberian Plateau (64°12′ N, 100°27′ E). This pure Gmelin larch forest (hereafter known as larch) was 105 years old, with mosses of *Pleurozium schreberi* and *Aulacomnium turgidum*, as well as the dwarf-shrubs of *Vaccinium vitis-idaea*, *V. uliginosum* and the shrub of *Betula nana* dominating the ground vegetation (Figure 2 left). Moss cover formed a thick porous layer of 10–25 cm in depth above the mineral soil and functioned as a heat insulator. The stand structure was characterized by a stem density of about 5480 living trees ha$^{-1}$, an LAI of 0.6 m$^2$ m$^{-2}$, a phytomass (dry weight) of 1.14 kg m$^{-2}$, a mean diameter of the trees at breast height of 6.5 cm and a mean tree height of 7.1 m. Individual tree crowns rarely overlap with one another [29].

The stand was located on a slightly sloping ancient rock terrace of the Nizhnyaya Tunguska River with a northern aspect. Soil type according to the US Classification System (Soil Survey Staff, 1998) [30] was Gelisols with a permafrost table existing within the upper 1 m of the soil profile [31]. The soil texture of the surface-active layer was clay rich; each soil horizon had a high rock fragment ratio.

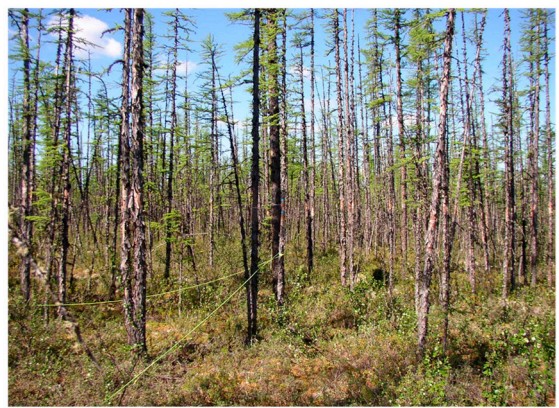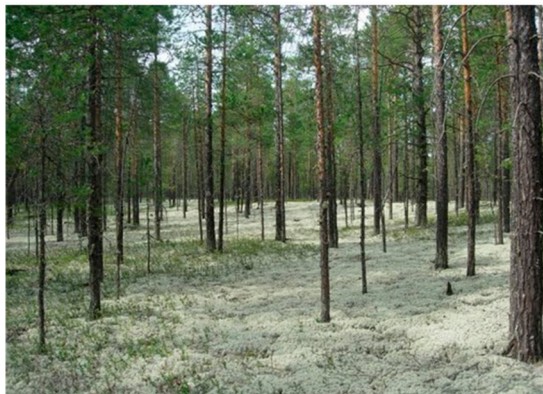

**Figure 2.** Study sites: the *Larix gmelinii* with *Pleurozium schreberi* and *Aulacomnium turgidum* and dwarf-shrubs forest floor (**left**) and the *Pinus sylvestris* with *Cladonia*, *Cladina*, and *Cetraria* forest floor (**right**).

The pine forest was located near the village of Zotino, about 30 km west of the Yenisei River at the eastern edge of the West Siberian Plain (60°44′ N, 89°09′ E). This forest was 215 years old, with lichens of the *Cladonia*, *Cladina*, and *Cetraria* species dominating the ground cover (Figure 2 right). The stand structure was relatively homogenous with a stem density of 468 living trees per ha$^{-1}$, a basal area of 30 m$^2$ ha$^{-1}$, an LAI of 1.5 m$^2$ m$^{-2}$, a phytomass (dry weight) of 10.7 kg m$^{-2}$, a mean diameter of the trees at breast height of 28 cm, a mean tree height of 23 m, the crown cover was 30%–60% [32].

The stand was located on slightly undulating, alluvial sandy soil with no underlying permafrost characterized according to the US Classification System [30] as a Pergelic Cryochrept [32]. The clay content in the mineral soil was less than 5%, and the soil was stone free. The layer of clay-rich material found at about 2.0 m depth was of potential importance for tree water supply in summer [33].

*2.2. Data Processing for Net Ecosystem Exchange of CO$_2$*

For recent decades the eddy covariance method has been used to monitor greenhouse gas (including CO$_2$) fluxes as the most common in meteorological and environmental studies [34]. The eddy covariance flux towers are currently distributed across multiple biomes of the world, with a special interest in the Arctic-boreal zone.

The fluxes are determined using the standardized equipment and data processing technique, which provides not only the high consistency of measured fluxes but also the representative global pattern of emission and uptake of greenhouse gases [35]. For details on instrumentation and data processing, see Supplementary File S2.

In Russia, the eddy-tower network for monitoring greenhouse gas fluxes is still sparse, and the towers are extremely inconsistently distributed over the territory. Specifically in Siberia, the regional and global carbon balance and their impact on the climate system remains most unexplored [22].

Net ecosystem CO$_2$ exchange (NEE) between the forest and the atmosphere is computed as:

$$NEE = \overline{w\prime c\prime} + \int_{o}^{h} \frac{\partial \overline{c}}{\partial t} dz \qquad (1)$$

where $h$ is the eddy measurement level, $w\prime$ is vertical wind velocity fluctuations, and $c\prime$ is CO$_2$ density fluctuations. The first term in Equation (1) is eddy covariance flux, and the second is storage change in CO$_2$ below the eddy flux measurement level. The method used by Hollinger et al. [36], calculating the change in CO$_2$ concentration at the top of the tower, was employed for estimating the second term in Equation (1).

Gaps in half-hourly NEE shorter than three half-hour periods were filled using linear interpolation. Longer gaps were filled with simple experimental regressions only for the purpose of integrating daily and seasonal NEE.

A quality check for our half-hourly eddy measurements based on energy balance closure showed that the imbalance varied between 19 and 24% in the pine site in 1998–2000 [33] and up to 34% in the larch site or 17% for the daily integrated [13]. On average, in eddy covariance studies, energy imbalance varies from several percent up to 40%, on average about 20% [37–40]. Among the main reasons for this imbalance are spatial landscape heterogeneity, incomplete detection of all turbulent and advective fluxes, and instrument malfunctions [40–42].

Half-hourly $CO_2$, water vapor and energy fluxes for the pine site processed in 1998–2000 (Carbon-Siberia, see Tellus 54 B 2002) and those for the Gmelin larch site processed in 2004–2008 (AsiaFlux, http://www.asiaflux.net (accessed on 6 February 2023)) were used in this study to calculate net ecosystem exchange (NEE), ecosystem respiration ($R_{eco}$) and gross primary production (GPP), and energy balance components.

$R_{eco}$ was calculated from air temperature using a $Q_{10}$-type equation [43]. Half-hourly night-time NEE under PAR < 10 μmol m$^{-2}$ s$^{-1}$ and friction velocity, u* > 0.2 m s$^{-1}$ were bin-averaged for the bin-width of the temperature of the air Ta = 1 °C. The relationship was of the exponential type:

$$NEE_{night} = k_0 \exp(k_1 T_a) \tag{2}$$

where $k_0$ and $k_1$ are fitting parameters and $T_a$ is air temperature.

Because the range of surface soil temperature in the larch forest did not even reach 10 °C during the vegetation season, we used the air temperature in Equation (2) to calculate $R_{10}$ and $Q_{10}$. $R_{10}$ was 1.1 μmol m$^{-2}$ s$^{-1}$, and the average $Q_{10}$ was 1.33 for the growing seasons in 2005–2008. In the pine forest, $R_{10}$ was 1.43 μmol m$^{-2}$ s$^{-1}$, and the average $Q_{10}$ was 2.35 during the vegetation season in 1998–2000.

Water use efficiency (WUE), by definition, is the amount of dry matter produced per unit of water transpired [44]. We evaluated WUE as follows:

$$GPP/E \tag{3}$$

where GPP was approximated as (NEE + $R_{eco}$) μmol $CO_2$ m$^{-2}$ s$^{-1}$ and E was approximated as ecosystem transpiration + evaporation measured above the forest. We calculated the daily GPP/E from May to October in dry 1999 and moist 2000 in the pine forest and from June to mid-September in dry 2005 and wet 2008 in the larch forest.

## 3. Results

### 3.1. Site General and Specific Climatology

Our pine and larch sites were located in contrasting environments, in which the permafrost and cold soils of the larch site seemed to create a specific soil climate that was critical to control the carbon exchange. The descriptive long-term climatology of the annual and growing seasons of both sites is given in Table S1 (Supplementary File S3) based on the data of two weather stations: Tura (64.3° N and 102° E), located 20 km west of the larch site, and Sym (60.3° N and 87.8° E) located 30 km west of the pine site. The distance between these weather stations was about 650 km. Climatic data were derived from Reference books on climate (1965–1970) [45]. We compared the pre-1960 means found in these Reference books to the means we calculated for the 1960–1990 period, which was accepted as a baseline period in climate change studies. We found that the January temperature increased by 2–3 °C, the July temperature increased by 0.5 °C, and that annual precipitation increased by 10% on average at both sites [46].

We compared the January and July temperature and annual precipitation means for the baseline period of 1960–1990 and the means for the 1991–2009 period. We also compared the means for the years of eddy-covariance measurements 1998–2000 (Zotino site) and 2004–2008 (Tura site) and the means for the 1991–2010 period to typify how representative

were the 3 and 4 years of our eddy-covariance measurements in two sites in the background of the two last decades (Figure S1, Supplementary File S3).

The 1991–2009 period became warmer in both summer by 0.7 °C and winter by 1.3 °C in Sym (the pine site), especially in the north by 2.6 °C in Tura (the larch site). Precipitations increased in both study sites: only 11 mm/year (2%) in Sym and significantly 30 mm/year (7%) in dryer Tura. As for the periods of our eddy-covariance measurements, in Sym, the 1998–2000 period was 5 °C colder in winter and 80 mm/year dryer than during the 1991–2009 period; however, the summer was 0.6 °C warmer. In Tura, the 2004–2008 winter was 1 °C warmer, and summer was 0.7 °C cooler than during the 1991–2009 period; precipitation was 45 mm/year less (Table S1 and Figure S1, Supplementary File S3).

The larch forest is in a cold, dry climate with very long, cold winters, which include a 40 cm snow pack. Summers are relatively warm but short, with low precipitation. The period with a positive temperature is 40% of the year span, of which 80% is the growing season which is accepted in ecological studies as the period when temperatures exceed the threshold of 5 °C. Cumulative degree days above 5 °C are only 845 °C, and negative degree days are extremely low, about −5000 °C. These climatic conditions result in a high continentality of climate characterized by a Conrad's index greater than 90. In turn, high continentality and low precipitation (slightly higher 300 mm $yr^{-1}$) support continuous permafrost and low thawing (active) layer depths (ALD).

Soils at the station Tura, under short grass cover, are not frozen at the surface during May–September, but stay frozen below a depth of 0.8 m prior to June and below a depth of 1.6 m prior to August and are never unfrozen at the 3.2 m depth. The pine forest exists in much milder conditions. Higher winter temperatures, longer growing seasons, warm soils with no permafrost, greater precipitation and a higher snow pack protecting soils from deep seasonal freezing and early thawing—all these climatic conditions favor the $CO_2$ exchange, higher carbon balance, and greater productivity of the pine forest in comparison to the larch forest.

Site-specific climatology in both sites during the growing seasons in 1998–2000 in the pine forest and in 2004–2008 (except 2006) in the Gmelin larch forest is shown in Figure 3.

Summer months were, on average, 1.5–2 °C warmer in the pine site. The principal habitat contrast between these sites was initiated by soil climate, specifically by the soil temperature driven by underlying permafrost. (Figure 3, middle and lower rows). In the larch forest with moss floor insulating heat fluxes, the surface 5–10 cm soil layer started thawing only in the second half of June, so larch leaf onset may start in the second half of May from our phenological observations under the frozen mineral part of the soil. Peak summer temperatures of the near-surface soil (5 cm depth) reached only 4–6 °C at the larch site compared with 14–17 °C at the pine site. Soils in the larch forest reached their maximum depth of thaw only by the second half of August when soil temperatures reached their maximal values at all depths in both sites. However, those values significantly varied: at 10 cm, they were 4 °C in the larch forest and 14 °C in the pine forest; at 30 cm—3 and 12 °C; at 50 cm—2 and 11 °C, and at 1 m—below 0 °C and 10 °C respectively. The ALD in the larch forest may be approximated as 80–90 cm from Figure 3 (the lower row) based on our field measurements of soil thawing. This range of soil temperatures created very contrasting conditions for ecosystem activities and especially soil respiration [47,48].

### 3.2. Net Radiation, Energy Balance, Bowen Ratio and Water Balance

Site-specific energy balance components are shown in Figure 4. Net radiation ($R_n$) was greater in the pine forest than in the larch forest due to a 2.5 weeks longer growing season and ~20% higher values of summer months, which resulted in totals $R_n$ = 1415 MJ $m^{-2}$ vs. $R_n$ = 1200 MJ $m^{-2}$ in the growing season respectively. Net radiation at our forest sites was compared to that at two actinometric stations, Eniseisk (near the pine forest) and Tura (near the larch forest), and showed their reasonable correspondence. The surface albedo difference of 5%–10% between the forest (at our sites) and grass (at weather stations) explained the difference in $R_n$ measured at our sites and the weather stations: the

darker forest surface absorbed more short-wave radiation than the lighter grass surface [49]. Maximal values of $R_n$ were ~120 W m$^{-2}$ above the larch forest and ~150 W m$^{-2}$ above the pine forest in June–July.

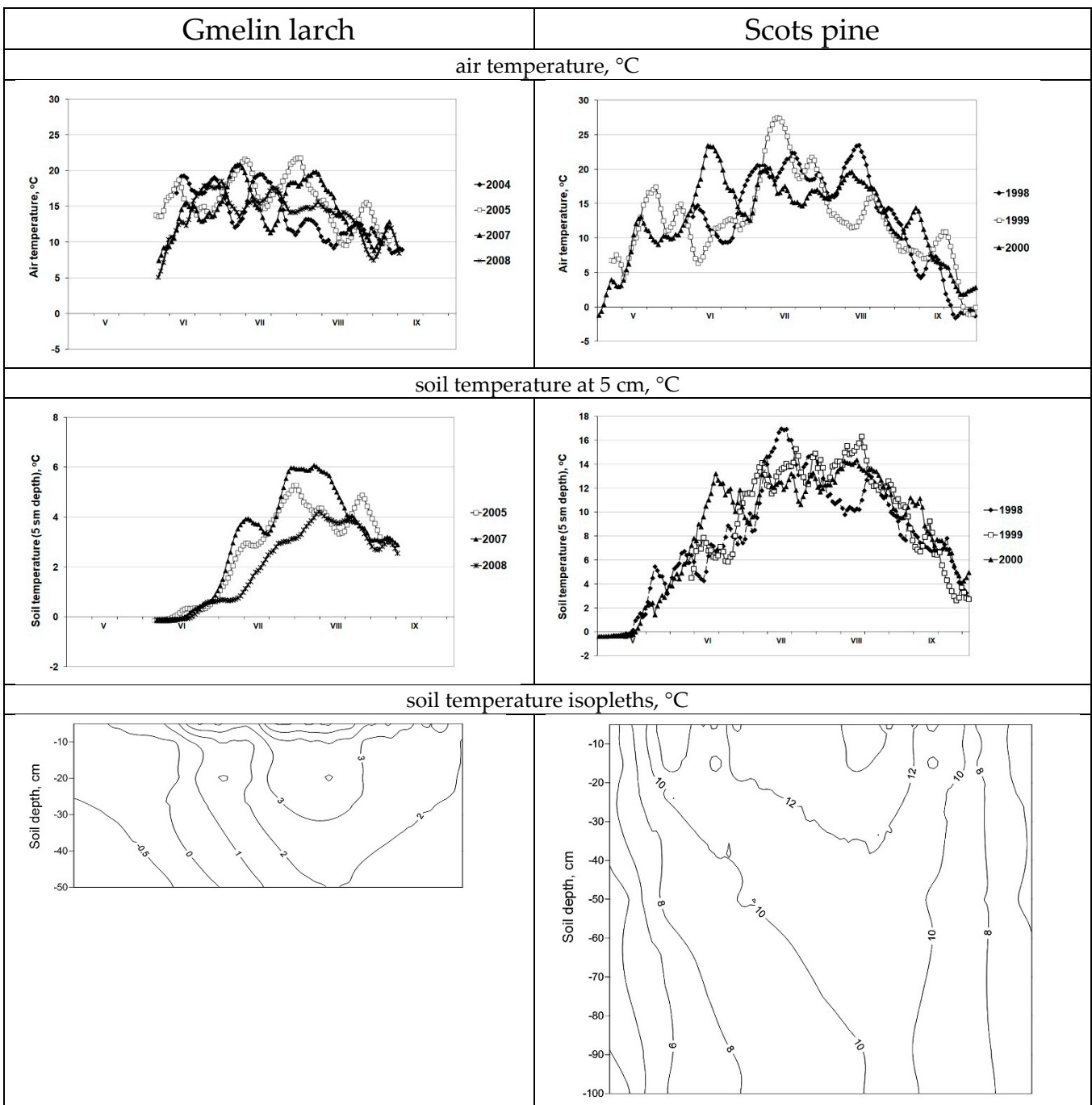

**Figure 3.** Temperature of the air (**upper**) and soils at 5 cm (**middle**) and soil temperature isopleths at various depths (**lower**) for one season in the Gmelin larch (**left**) and Scots pine (**right**) ecosystems for the growing season.

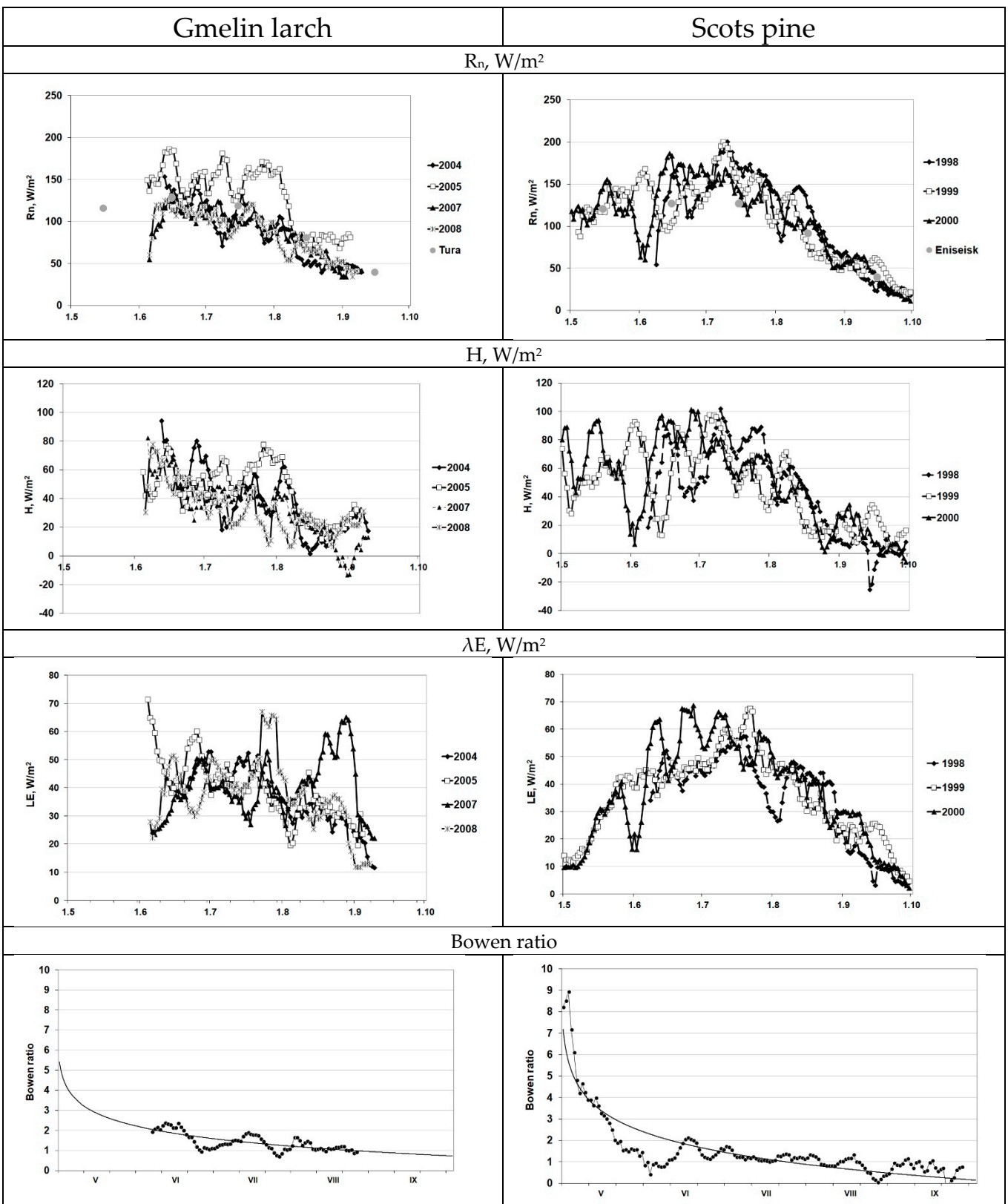

**Figure 4.** Energy balance components and Bowen ratio (positive 7-day running averages) in the Gmelin larch (**left**) and Scots pine (**right**) ecosystems for the growing season (May–September, V–IX): $R_n$–Net radiation (filled dots are data from actinometric stations Tura (near the Gmelin larch forest) and Eniseisk (near the Scots pine forest); H—sensible heat, and $\lambda E$—Latent heat; Bowen ratio.

Partitioning of the radiation balance into sensible heat flux (H) and latent heat flux (LE) expressed through both absolute values and Bowen ratio ($\beta$ = H/LE) in both forests are shown in Figure 4.

Soil heat flux (G) in the non-permafrost pine forest was ~3%–5% of $R_n$ [33]. G was not measured in our larch forest, but it was expected to be much greater because of permafrost thawing. Pavlov (1984) [50] estimated soil heat flux could reach 30% of $R_n$ during the thawing process in spring and 50% in fall.

Sensible heat flux was the larger term in the radiation balance and followed the $R_n$ pattern in both forests. In the Gmelin larch forest, H was maximal 40–80 W m$^{-2}$ at the beginning of the growing season, in late May–early June, before leaves came out and after decreased continuously. In the pine forest, with its evergreen habit, H stayed high till August and was 60–100 W m$^{-2}$.

Latent heat flux was governed by both available energy and rain. Maximal LE reached 40–60 W m$^{-2}$ also in the mid-summer in both ecosystems. All LE extremes (maxima and minima) were related to flash rain events, such as in July of 2007 and in August of 2008 in the larch forest or little rain in the pine forest in July of 1998 (Figures 3 and 4).

Bowen ratio. In both our pine and larch forests, $\beta$ stayed between 1 and 2 during the growing season and only in early spring was high and reached 10 (Figure 4). Compared to our pine forest, in a closely located but moister spruce-fir forest and a mixed birch-conifer forest $\beta$ was ~1 and in a birch forest $\beta$ < 1 during the growing season (Röser et al. 2002) [51]. Dolman et al. (2004) [10] also indicated that in their Cajander larch forest in central Yakutia, $\beta$ remained 1–2 when the forest was physiologically active and raised to 10 when it was not. Ohta et al. [52] showed that in a sparse Cajander larch forest also in Yakutia $\beta$ varied between 10–20 in early spring before leaves came out and then dropped to 1 during the full foliage in the mid-summer and raised again in the fall. Pavlov (1984) [50], though, gave lower $\beta$ < 1 in both pine and larch forests in central Yakutia and $\beta$ > 1 only in early spring.

Water balance. Precipitation (P), evaporation (E, calculated as LE/L), and the difference between them (P–E) (without considering water from thawed permafrost) for the growing season are shown in Figure 5 and Table 1.

**Table 1.** Water balance: precipitation (P), evaporation (E), and (P-E) for the growing season (May–September) in the Gmelin larch and pine ecosystems.

| | Larch Ecosystem | | | Pine Ecosystem | | |
|---|---|---|---|---|---|---|
| | **2005** | **2007** | **2008** | **1998** | **1999** | **2000** |
| P | 122 | 184 | 215 | 248 | 233 | 228 |
| E | 112 | 127 | 116 | 173 | 192 | 200 |
| P–E | 10 | 57 | 99 | 75 | 41 | 28 |

Precipitation was, on average, ~20% greater at the pine site for the growing season. Some extreme rain events of 100 and 120 mm in the larch site in the wet July of 2008 and August of 2007 made some short-term exceptions to the general rule, though. Those monthly precipitation amounts significantly exceeded any monthly rain observed in 1998–2000 at the pine site. However, in the pine forest, some dry months, such as July of 1998, when only 10 mm of rain occurred, were never observed at the larch site (Figure 5). On average, the 3-year water balance was positive during the growing season in both the pine and larch forests. However, evaporation was 90% of precipitation during the dry growing seasons of 1999 (pine forest) and of 2005 (larch forest) (Table 1). Although the water balance was positive for the growing season, the monthly and cumulative daily water balance were often negative (Figures 4 and 5). This negative water balance was likely compensated by soil water available from a deeply located water table in the pine forest [33] and from thawed permafrost water [7] in the larch forest. In the wet growing seasons of 2000 (pine forest) and 2008 (larch forest), evaporation was about 50%–70% of rain (Table 1) because flash-rainwater was quickly discharged after rain events and was not available for evaporating. During wet growing seasons, water balance was positive,

accumulating ~80 mm and 100 mm of rain by the end of the season in the pine forest and in the larch forest, respectively (Figure 5).

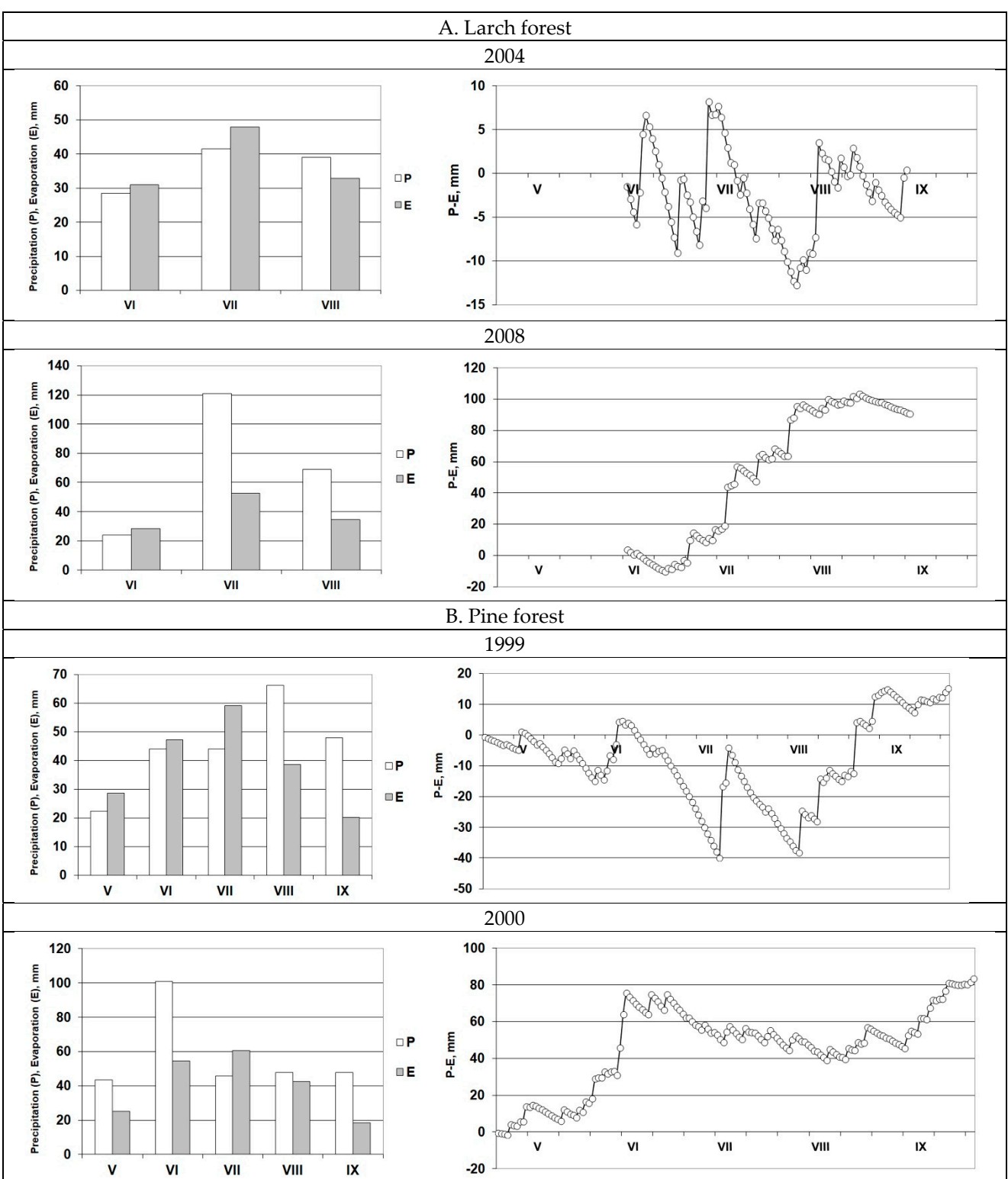

**Figure 5.** Monthly precipitation (P) and evaporation (E) (**left**) and cumulative water balance (P–E) for the growing seasons (**right**) in the dry year (2004) and in the wet year (2008) in the Gmelin larch and (1999) and (2000) respectively in the Scots pine ecosystems.

### 3.3. Daily, Seasonal, and Annual Dynamics of Ecosystem $CO_2$ Exchange: NEE, $R_{eco}$, and GPP

The daily dynamics of the ecosystem $CO_2$-exchange have similar patterns in our larch and pine stands over all observational years, although their curve shapes were different due to day lengths and NEE values: more compressed in NEE and prolonged in the day span in the larch pattern and, conversely, extended and shortened in the pine pattern (Figure 6 upper). Night-time respiration turned to day-time $CO_2$ assimilation at 3–9 a.m. in the larch forest compared to 6–8 a.m. in the pine forest. In the summer, photosynthetic activities (PAR > 10 μmol m$^{-2}$ s$^{-1}$) in the larch forest lasted longer than in the pine forest due to longer light days at the 64° N latitude vs. 60° N latitude (the «polar day» effect): 16–18 h vs. 14–15 h a day.

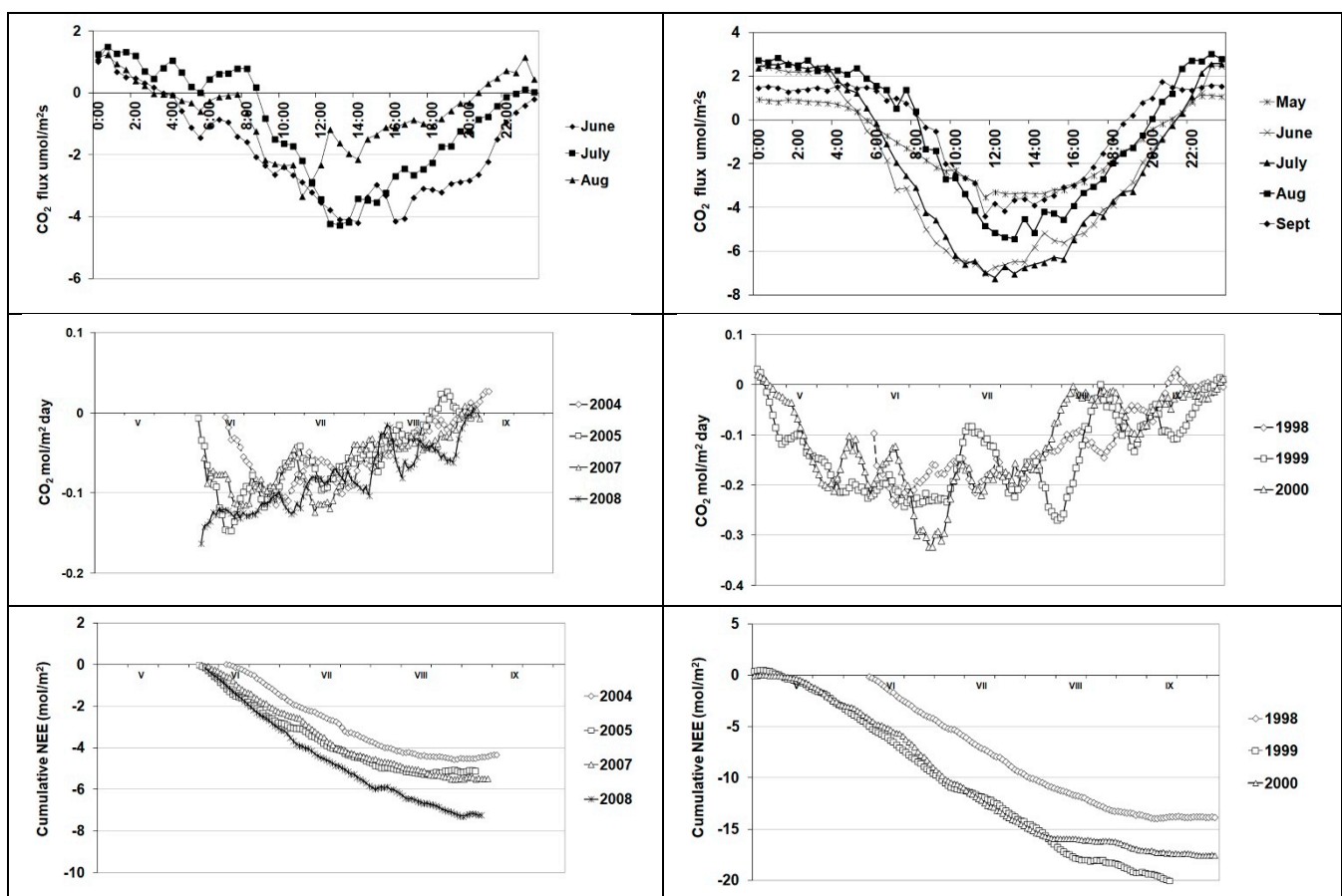

**Figure 6.** Daily (**upper**), seasonal (**middle**) and cumulative (**lower**) NEE in the Gmelin larch (**left**) and Scots pine (**right**) ecosystems for the growing season (May–September, V–IX).

In mid-July, daily maximal half-hourly $CO_2$ flux reached $-$(9–11) μmol m$^{-2}$ s$^{-1}$ in the larch forest and $-$(10–12) μmol m$^{-2}$ s$^{-1}$ in the pine forest (not shown). However, the average daily maximum $CO_2$ uptake in June and July was smaller in the larch ecosystem (4 μmol m$^{-2}$ s$^{-1}$) than in the pine forest (7 μmol m$^{-2}$ s$^{-1}$, Figure 6 upper). Thus, a longer day effect in the larch forest did not yield a larger daily total $CO_2$ uptake.

In the mid-summer, June–July, NEE crossed the null at midnight, becoming a $CO_2$ source and at 4 am, becoming a $CO_2$-sink in the larch forest (Figure 6 upper). In early September, when the needles started yellowing, the daily total NEE became about a null or changed to a positive sign (Figure 6 middle). At the summer peak, the pine forest became a $CO_2$ source from 9 p.m. to 6 a.m., which was 5 h longer than the larch forest was a $CO_2$ source. In October, NEE became positive and stayed such till April. In general, daily $CO_2$ exchange rates were about twice as large in the pine forest than in the larch forest (Figure 6 middle). In the larch ecosystem, NEE was $-0.1$ mol $CO_2$ m$^{-2}$ day$^{-1}$ at the beginning of the

vegetation season in June, reached its maximum, $-0.22$ mol m$^{-2}$ day$^{-1}$, in July, and crossed null again at the end of August. The pine ecosystem performed a greater NEE during the growing season, starting at 0–0.2 mol m$^{-2}$ d$^{-1}$ in May, reaching 0.2–0.3 mol m$^{-2}$ day$^{-1}$ in midsummer, and crossing null again at the end of September.

In the Gmelin larch forest, seasonal NEP (-NEE integrated over a long period [39] accumulated for the growing season varied from 63 to 106 g C m$^{-2}$, with an average of ~83 g C m$^{-2}$ season$^{-1}$ for 2004–2008 (Tables 2 and 3). Assuming that winter respiration in the cold environment of interior Siberia was low, we concluded that both ecosystems stayed a C-sink year-round of different strength, though. The seasonal NEP in our Gmelin larch ecosystem appeared to be the weakest among Siberian boreal ecosystems and other boreal forest ecosystems reported in the literature, except one that was also on permafrost in central Yakutia [53]. If to conclude from extremely low maximum net $CO_2$ uptake reaching ~5 µmol m$^{-2}$s$^{-1}$ and ~0.1 mol m$^{-2}$d$^{-1}$ in the mid-summer, a similar Gmelin larch forest in East Siberia [53] looks very much of the same metabolism and grows in a similar environment to our Gmelin larch forest located in Central Siberia. In the pine forest, NEP was between 212 and 251 g C m$^{-2}$ season$^{-1}$, with a 3-year average of 228 g C m$^{-2}$ season$^{-1}$ (Figure 6 lower, Table 2). The NEP patterns for all observation years showed a strong signal of both studied ecosystems being a C-sink for the growing season. On average, in West and Central Siberia, NEP in middle taiga pine forests was measured by inventory methods as 160–225 g C m$^{-2}$ yr$^{-1}$ [54].

**Table 2.** NEP, $R_{eco}$ and GPP for the growing season in the Gmelin larch and pine ecosystems.

| | Larch Ecosystem | | | Pine Ecosystem | | |
|---|---|---|---|---|---|---|
| | **2005** | **2007** | **2008** | **1998** | **1999** | **2000** |
| GPP, gCm$^{-1}$ | 167 | 186 | 210 | 543 | 592 | 552 |
| $R_{eco}$, gCm$^{-1}$ | 104 | 106 | 104 | 324 | 341 | 340 |
| NEP, gCm$^{-1}$ | 63 | 80 | 106 | 219 | 251 | 212 |

May of 1998 was approximated by the May average of 1999–2000.

**Table 3.** Relationships between $CO_2$ exchange and energy budget components for the growing season in the Gmelin larch and pine ecosystems.

| Ecosystem | | $R^2$ ($p < 0.001$) | | |
|---|---|---|---|---|
| | | **H** | **LE** | **R$_o$** |
| Larch | 2004 | $-0.5$ | 0.8 | 0.7 |
| | 2005 | $-0.6$ | 0.4 | 0.8 |
| | 2007 | $-0.7$ | 0.3 | 0.8 |
| | 2008 | $-0.8$ | 0.4 | 0.8 |
| Pine | 1998 | $-0.9$ | 0.9 | 0.9 |
| | 1999 | $-0.6$ | 0.7 | 0.7 |
| | 2000 | $-0.6$ | 0.7 | 0.7 |

Cumulative ecosystem respiration for the growing season in the Gmelin larch forest was 106 g C m$^{-2}$ on average and did not change from year to year. In the pine forest, it was 3 times greater—335 g C m$^{-2}$. In both our forests, $R_{eco}$ appeared to be lower, especially in the larch forest, compared to 25 various forest ecosystems in West Europe [55] and to 8 ecosystems in Northern Asia [56] that differed by composition. Kononov (2006) [57] calculated $R_{eco}$ = 400 g m$^{-2}$ for the 2001 growing season in a Cajander larch forest in Yakutia on permafrost (with a deep active layer, though).

$R_{eco}$ summed to NEP yielded 167–210 g C m$^{-2}$ of gross primary production (GPP) for the growing season in the larch forest and 542–592 g C m$^{-2}$ in the pine forest (Table 2). Compared to the GPP of the Cajander larch forest [10], the GPP of our Gmelin larch forest

was 2-fold lower. In the pine forest, with regarding winter respiration ~65 g C m$^{-2}$ as measured in the pine forest for the 1999–2000 winter [33] and GPP ~560 g C m$^{-2}$ season$^{-1}$, then we estimated annual GPP ~625 g C m$^{-2}$ for the growing period. Lloyd et al. [58] estimated annual GPP as 627 g m$^{-2}$ of our pine forest using "maximum data" of the $R_{eco}$ reported by Shibistova et al. [47].

### 3.4. NEE, Heat and Water Vapor Fluxes

We expected that there would be a relationship between $CO_2$ exchange and energy and water budget components because stomata activities are directly related to heat and water accessibility. We expected, therefore, that correlations between $CO_2$ exchange and evapotranspiration in the dry climate of interior Siberia would be particularly significant. These relationships were complex and not readily apparent during any of the entire growing season; two-thirds of the determination coefficients were, on average, rather high, $R^2 > 0.5$, explaining at least 50% of the daily NEE variation (Tables 2 and 3).

In the dry larch forest with limited moisture, LE explained 70% of the NEE variation in the dry summer of 2004, while in the wet summers of 2007 and 2008, when it was likely that evaporation was unlimited, it explained only 10–20%. In the pine forest, LE explained 80% of the NEE variation in the dry summer of 1998. Even in wet summers, heat components $R_n$ and H explained only half of the NEE variation in both forests. This fact confirmed the assumption that in the dry climate of interior Siberia, water primarily governs ecosystem processes.

In addition, the non-linear (a parabolic-type) relationship between NEE and the Bowen ratio (β) was established in both ecosystems with $R^2 = 0.8$ ($p < 0.001$) in the pine forest and $R^2 = 0.3$ ($p < 0.001$) in the larch forest (Figure 7). The maximal $CO_2$ exchange, $-(200–270)$ mol d$^{-1}$ in the pine forest and $-(130–200)$ mol d$^{-1}$ in the larch forest, were related to a range of β ≈ 1–2.5. Under lower β, β < 1, cool and wet conditions, $CO_2$ exchange decreased in both ecosystems and could become positive, turning the system into a $CO_2$ source. Under high β, β > 2.5, dry and hot conditions, $CO_2$-exchange also decreased, likely because stomata closed under high sensible heat and water vapor deficit which are trade-offs between water loss and C gain in the process of photosynthesis.

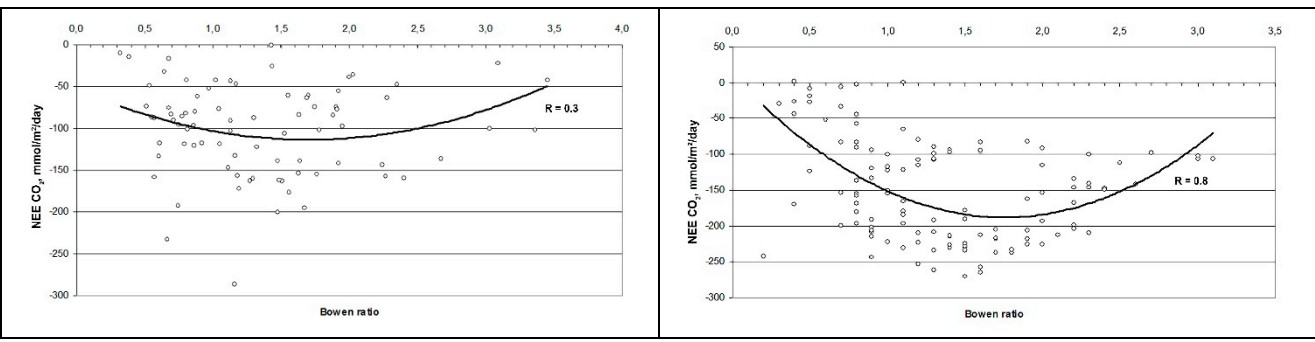

**Figure 7.** The relationship between NEE and Bowen ratio in the Gmelin larch (**left**) and Scots pine (**right**) ecosystems for the growing season.

### 3.5. Water Use Efficiency

Both GPP and E in our ecosystems had a distinct seasonal course with maxima in the mid-summer. Both GPP and E were greater in the pine forest due to a longer growing season and higher rainfall, although water from thawed permafrost in addition to rain was available in the larch forest. However, the bulk canopy WUE was constant in both ecosystems with no obvious seasonal course because low values of WUE did not occur during our observation period when GPP dropped to null in the dormancy period. In each figure, we added monthly means of WUE (black dots) assuming that WUE = 0 in months with no physiological activities under a mean air temperature <0 °C: from 10 October to 26 April in the pine forest and from 2 October to 7 May in the larch forest (Table S1,

Supplementary File S3). WUE varied between 1 and 3 $\mu$mol $CO_2$ mmol$^{-1}$ $H_2O$ in the larch and 2 and 5 $\mu$mol $CO_2$ mmol$^{-1}$ $H_2O$ in the pine forests. WUE was not much different in wet and dry years (Figure 8). On a seasonal basis, a 3 year average WUE was 2.3 in the larch forest and 4.4 $\mu$mol $CO_2$ mmol$^{-1}$ $H_2O$ in the pine forest or 5.8 mg $CO_2$ g$^{-1}$$H_2O$ and 11 mg $CO_2$ g$^{-1}$$H_2O$, respectively. The reciprocal of WUE is the water cost per unit C assimilation at a time scale. The water cost per unit C-assimilation was 170 g $H_2O$ in the pine forest vs. 260 g $H_2O$—in the larch forest.

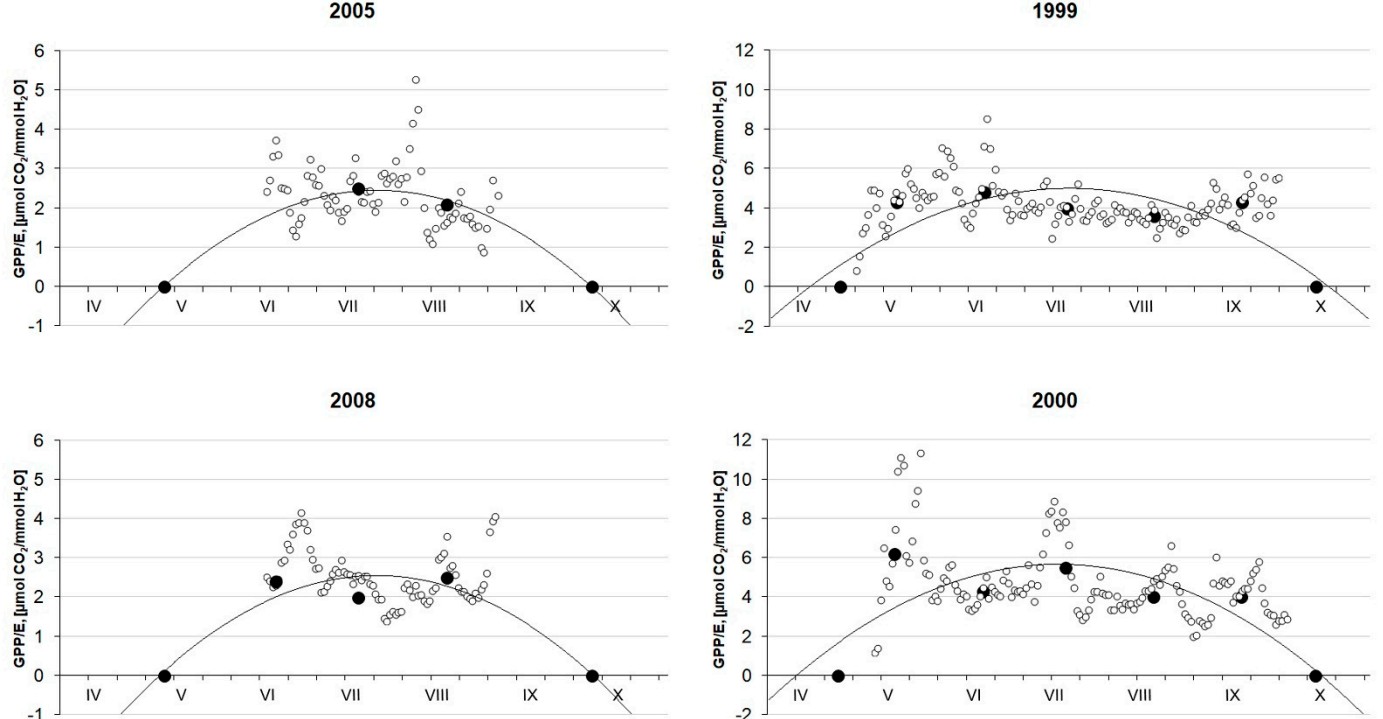

**Figure 8.** Water use efficiency (GPP/E) in dry (1999 and 2005) and wet (2000 and 2008) years in the Gmelin larch forest (**left**) and Scots pine forest (**right**) during May—September (V–IX).

## 4. Discussion

Interior Siberia is noteworthy from the viewpoint of the conifer species that sharply changes from evergreen conifers (*Pinus sibirica*, *P. sylvestris*, *Abies sibirica*, *Picea obovata*) dominating west of the Yenisei River to deciduous conifers (*Larix gmelinii* and *L. cajanderi*) east of the Yenisei [59]. *L. gmelinii* and *L. cajanderi* form the largest larch biome on the Earth, distributed on the continuous permafrost of Northeastern Eurasia. *Larix sibirica* grows only on non-permafrost and may be admixed to evergreen conifers (see for details in Supplementary File S1).

The larch forest that was located 4 degrees north of the pine forest, and therefore the accumulated daylight over the growing season is 200 h greater than daylight in the pine forest, which yielded about two additional weeks for photosynthesis. However, due to a 2.5 weeks shorter growing season in the north, the larch forest likely lost its only advantage to accumulate higher NEP and GPP.

We compared the direct functional parameters of our two forests, the permafrost Gmelin larch forest at Tura and the non-permafrost Scots pine forest at Zotino, with findings in the same forests from recent publications. Olchev et al. [17] studied energy fluxes and NEE in the Gmelin larch forest; Masyagina et al. [18] studied soil respiration in both Scots pine and Gmelin larch forests; and Urban et al. [19,20] studied transpiration and water use efficiency in Scots pine, Siberian larch and Gmelin larch in the warm subtaiga forests. We found that these ecosystem parameters during the growing season remained very similar after a decade or so: NEP = (76.5–103.6) g C m$^{-2}$ in 2013–2015 [17] vs. (63–106) g (this study)

in the permafrost larch forest; SR = 310 g C m$^{-2}$ in the pine forest in 1995–2010 [18] and 335 g C m$^{-2}$ (this study); SR = 217 g C m$^{-2}$ in the permafrost larch forests 1995–2010 [18] vs. 105 g C m$^{-2}$ (this study).

Average $CO_2$ uptake in our permafrost larch forest, $-(3$–$6$ μmol m$^{-2}$ s$^{-1})$, was smaller than $-(6$–$13$ μmol m$^{-2}$ s$^{-1})$ in non-permafrost boreal sites around the world: in North America, a jack pine forest [60], a black spruce forest [61]; in Central Siberia, fir-spruce or fir-birch forests [51]; in Northern Finland, a subarctic birch forest [62]. Compared to other larch forests on continuous permafrost, the maximal mid-summer net uptake in our Gmelin larch forest was close to that of an *L. gmelinii* forest in eastern Siberia $-5$ μmol m$^{-2}$ s$^{-1}$ [53] and was much smaller than reported for an *L. cajanderi* forest, $-(10$–$18)$ μmol m$^{-2}$ s$^{-1}$, in the "Spasskaya Pad" site near Yakutsk, also eastern Siberia [10]. The maximal net uptake in this forest was comparable to that of a productive non-permafrost *L. sibirica* forest in Mongolia [63].

The daily uptake rates in our larch forest during mid-summer and, thus, the cumulative NEE during the growing season were considerably lower than in other mature boreal forests, including the same species of larch reported in the literature. This may be due to the smaller LAI (<0.6) of our Gmelin larch forest compared to other forests (e.g., 1–4 at the BOREAS study sites in Canada [60,61], [64] and that of the Cajander larch forest near Yakutsk with an LAI of ~2 [10]. Another crucial reason for the large differences in NEE between various boreal forests, including the Cajander larch forest near Yakutsk and our Gmelin larch near Tura, was the low active layer depth of 0.8 m and cold soils. ALD was 1.2–1.4 m [10] and reached 1.8 m [1] in Cajander larch forests on warm sandy soils in Yakutia within the permafrost zone.

The physiological activity of larch trees in the permafrost ecosystem followed the soil temperature increase in the upper soil horizon. A positive temperature of ~1–3 °C in the litter layer, where most larch fine roots expand, commenced the growth activity [65]. In the northern taiga, this usually happened during the last week/10 days of May, sometimes at the beginning of June, when air temperatures reached +10 °C and initiated thawing in tree-less openings. Simultaneously, the buds of the larch trees began to burst, revealing new larch needles, when photosynthetic active radiation, PAR, reached 600–700 μmol m$^{-2}$s$^{-1}$ [15]. Thus, in early June, the synchronized influence of air and soil temperatures, as well as of PAR, triggered the processes of $CO_2$ exchange. However, a very weak net $CO_2$ uptake of $-1$ μmol m$^{-2}$s$^{-1}$ was observed in early summer.

Vygodskaya et al. [66] showed that in the Gmelin larch forest on permafrost in Yakutia, the maximum assimilation rate was reduced by 12% by low PAR and was reduced by 75% by low air humidity and water supply, including water uptake from thawing permafrost that is temperature limited. The weather conditions in our Gmelin larch site were characterized by the dry air and low soil moisture that explain the low assimilation activity of the forest. At the beginning of June, the process of the cambial division starts, the intensity of which resulted in the wood cell size and, finally, in wood stem productivity [67]. Orlov and Koshelkov [68] showed that the temperature threshold of 5 °C in pine non-permafrost forests is important for root physiological activities. The combination of temperatures below 5 °C with high photosynthetically active radiation, which often occurs in interior Siberia, may cause inhibited conifer photosynthesis in early summer [69].

Water use efficiency was 5.8 in the larch forest and 11 mg $CO_2$ g$^{-1}$H$_2$O in the pine forest. Thus, the water cost per unit C-assimilation was 1.5-fold greater in the larch ecosystem: 170 g vs. 260 g. This may be likely explained that a portion of water does not play a part in C-assimilation. Cold water drains along the frost table in the Gmelin larch ecosystem rather than it is consumed by roots unless water warms up during the summer. WUE of our Gmelin larch forest in central Siberia and that of a Gmelin larch forest in eastern Siberia [53] were similar. Hollinger et al. [53] also found that the model and independent leaf-level measurements suggest that the marginal water cost of plant C-gain in the *L. gmelinii* forest in East Siberia is very much similar to values from deciduous or desert species rather than other boreal forests. *L. sibirica* forests growing in the cold

semiarid environments in Mongolia at the southern limit of the Siberian taiga also exhibit a high water use efficiency [63].

WUE of both our ecosystems appeared to be consistent with those reported in the literature for a wide range of trees: 2.4–19.8 mg g$^{-1}$ [70]. Compared to some published results on WUE in temperate and boreal forests, our estimates were in this range: 8.5 mg $CO_2$ g$^{-1}$$H_2O$ in a temperate mixed broadleaved-conifer (Mongolian oak and Korean pine) forest in NE China [70]; 11.3 mg g$^{-1}$ in a boreal mixed conifer (Norway spruce and pine) forest in Sweden [71]; 10–21 mg g$^{-1}$ in a boreal willow forest in Sweden [72].

Based on our findings, in "hot spots" where Gmelin larch forests on permafrost were predicted to be replaced by the Scots pine forest [73] that follow the permafrost retreat in a warming climate, C-sink would increase uptaking $CO_2$ from the atmosphere. Thus, positive potential feedback to the climate system that slows down climate warming may take place. However, as predicted with general circulation models, profound warming is expected at high latitudes. Thus Arctic-boreal ecosystems become especially vulnerable to warming, which may transform the Arctic-boreal ecosystems from the carbon sink to source [27,28]. Abbot et al. [28] analyzed 98 permafrost-region experts' evaluations of biomass, wildfire, and hydrologic carbon flux responses to climate change. Experts assessed that the permafrost region will become a carbon source in the atmosphere by 2100.

## 5. Conclusions

- Two different forest ecosystems growing in contrasting habitats in interior Siberia were studied: a *Pinus sylvestris* forest growing on warm sandy soils and a *Larix gemilii* forest growing on permafrost soils with a shallow active layer depth. These forest ecosystems differ distinctively in their structure (age, height and diameter, LAI, stem density, etc.). The permafrost plays a double role: on the one hand, it supports the forest existence in a dry climate over East Siberia, delivering additional water from thawing permafrost; and on the other hand, much available energy, up to 30%–50%, is consumed in thawing ice. Thus less energy remains for sensible heat and latent heat flux, warming the soil and ambient air and for physiological processes in ecosystems.

- Net radiation was 2–2.5 fold greater in the pine forest than in the larch forest due to a 2.5 week longer growing season. Sensible and latent heat partitioned from $R_n$ and expressed by the Bowen ratio showed that β remained at 1–2 for the growing season when the pine forest was physiologically active and increased up to 8–10 when it was not.

- Precipitation and evaporation in the pine forest were 30%–50% greater than in the larch forest. In both ecosystems, the water balance was positive for the growing season; however, the monthly and cumulative daily water balances were often negative.

- Daily maximal half-hourly $CO_2$ fluxes were about the same in both ecosystems $\sim$−10 µmol m$^{-2}$ s$^{-1}$. However, averaged daily $CO_2$ fluxes in the pine forest were three times larger than the fluxes in the larch forest, which resulted in 228 g C m$^{-2}$ season$^{-1}$ vs. 83 g C m$^{-2}$ season$^{-1}$, respectively. The NEP patterns in both ecosystems exposed a strong signal of them being a C-sink for the growing season and year-round. Both $R_{eco}$ and GPP were 2–3 fold lower in our Gmelin larch.

- Water use efficiency (GPP/E) of the pine ecosystem appeared to be, on average, 2 times greater: 11 vs. 6 mg $CO_2$ g$^{-1}$$H_2O$ in the larch ecosystem. Thus the water cost per unit of C-assimilation was twice greater in the permafrost larch ecosystem.

**Supplementary Materials:** The following supporting information can be downloaded at: https://www.mdpi.com/article/10.3390/f14020346/s1, Supplementary: Suppl. S1.; Suppl. S2.; Suppl. S3.: Table S1. General climatology for the growing season (May–September) of two study sites.; Figure S1. Time series of annual precipitation (upper), July temperature (middle) and January temperature (lower) averaged for the 1960–1990 and 1991–2009 periods for two weather station Sym (upper in each figure) and Tura (lower in each figure). Means for both periods are horizontal and means for the periods of eddy-covariance observations are circled.; Suppl. S4.: Conclusions.

**Author Contributions:** Conceptualization, N.M.T.; methodology, N.M.T. and V.I.Z.; software, V.I.Z.; validation, N.M.T. and V.I.Z.; formal analysis, V.I.Z. and N.M.T.; investigation, N.M.T., V.I.Z. and O.A.Z.; data curation: V.I.Z., N.M.T., T.K. and Y.M.; writing—original draft preparation, N.M.T., V.I.Z. and O.A.Z.; writing—review and editing, N.M.T., V.I.Z., O.A.Z., E.I.P., T.K. and Y.M.; visualization, V.I.Z., O.A.Z., N.M.T. and E.I.P.; funding acquisition, N.M.T. All authors have read and agreed to the published version of the manuscript.

**Funding:** This research received no external funding.

**Institutional Review Board Statement:** Not applicable.

**Informed Consent Statement:** Not applicable.

**Data Availability Statement:** Not applicable.

**Acknowledgments:** This article is dedicated to the memory of Natalya N. Vygodskaya of the Ecology and Evolutionary Problems Institute, Russian Academy of Sciences, Moscow, Russian Federation, and Yuichiro Nakai of the Forestry and Forest Products Research Institute, Tsukuba, Japan, who organized eddy covariance measurements and supervised our carbon dioxide flux research in the pine forest and larch forest in central Siberia respectively. We sincerely thank Olga Gavrichkova for her internal review and Jane Bradford for English editing.

**Conflicts of Interest:** The authors declare no conflict of interest.

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
