# Peer review of "A Comparative Study of Climatology, Energy and Mass Exchange in Two Forests on Contrasting Habitats in Central Siberia: Permafrost Larix gmelinii vs. Permafrost-Free Pinus sylvestris"

_forests, doi:10.3390/f14020346_

Round 1

Reviewer 1 Report

This study studied the carbon dioxide flux and water use efficiency of two different age groups of forest types, which has important reference significance for the current study of forest carbon neutralization function. However, the manuscript also has the following problems:

1. The article is too long, so it is recommended to compress it substantially;

2. The introduction of the article is relatively short, and does not fully describe the research status at home and abroad, and the cited literature is not the latest. It is suggested to comprehensively elaborate the core content of the manuscript, summarize the innovation of this article and solve the scientific problems;

3. The experimental design was inadequate, and the differences between different places, including water, soil, tree growth and other information, were not clarified; The sampling time of the instrument and the method of data processing were not clearly stated;

4. It is suggested to move all parts 3.1 and 3.2 of the manuscript to the method part. This is the situation of meteorological factors in the study area, and it is not the result of the study; Move to the method section, and delete a lot, just keep the results;

5. The discussion on the manuscript is lack of pertinence, and the literature cited for comparative analysis is too old. It is recommended to conduct a systematic discussion on the 3.4, 3.5 and 3.6 parts of the manuscript; This part needs to be added, and the existing content is insufficient;

6. The conclusion of the manuscript is too long, just give the regular result directly; Proper compression is recommended;

7. The document citation format of the manuscript does not conform to the requirements of the journal.

Author Response

Rev.1

Comments and Suggestions for Authors

This study studied the carbon dioxide flux and water use efficiency of two different age groups of forest types, which has important reference significance for the current study of forest carbon neutralization function. However, the manuscript also has the following problems:

Dear Reviewer, thank you for your thorough reading and valuable comments. We fully accept your comments and will do our best to respond them.

  1. The article is too long, so it is recommended to compress it substantially;

Our article may seem too long because the idea to fulfill such a comparative study on energy and mass exchange and GPP/NEP in two contrasting forests over continental Siberia arose and was built up for several years in the view of how the permafrost may impact site climates controlling fluxes and ecosystem exchange in a changing climate. Similar comparative studies in Russia and Siberia in particular are rare. In Introduction and Discussion, we refer most recent publications for Russia: Olchev et al (2022) that studied energy fluxes (net radiation, latent heat and sensible heat); Masyagina et al (2020) studied soil respiration; and Urban et al. (2019; 2018) studied transpiration  and water use efficiency in the same Gmelin larch and Scots pine forests. From the summaries of Virkkala et al. (2022) and Natali et al. (2019a, b), we are aware of three forest eddy tower sites across Siberia: one non-permafrost site Zotino (in our study), two permafrost sites Tura (in our study) and Yakutia. A permafrost larch forest in Yakutia (Spasskaya Pad) was discussed in multiple publications of Dolman et al., Machimura et al., and Maximov et al. (see our References). None of those publications compared permafrost forest ecosystems to non-permafrost forest ecosystems in detail. In our present study, we suggested the analyses of multiple functional parameters (heat and water vapor fluxes (LE, H, β ); CO2 exchange (NEE, Reco, and GPP); and water use efficiency (WUE) of two principal forest ecosystems as Scots pine and Gmelin larch forests in interior Siberia.

Recently, Arctic-boreal landscapes, including both permafrost and non-permafrost ecosystems, have been intensively studied based on eddy covariance measurements collected from many eddy sites across the circumboreal zone (Ueyama M et al, 2013; Baldochhi et al. 2018; Natali et al. 2019 a, b; Virkkala et al, 2022; Watts et al, 2023) As predicted from general circulation models, profound warming is expected at high latitudes, thus Arctic-boreal ecosystems become especially vulnerable to warming which may transform the Arctic-boreal ecosystems from the carbon sink to source (Watts 2023).  Abbot et al. (2016) analyzed 98 permafrost-region experts’ evaluations of biomass, wildfire, and hydrologic carbon flux responses to climate change. Experts assessed that the permafrost region will become a carbon source to the atmosphere by 2100.

We compared directly these Siberian ecosystem parameters and found them to be very similar after a decade or so.

We installed eddy towers and started eddy-covariance measurements in a pine forest in 1998 (Tchebakova et al., 2002) and in a Gmelin larch forest in 2004 (Nakai et al., 2008).  Eddy measurements were available for our study during 1998-2000 and 2004-2008 respectively. Since then the eddy covariance measurements were interrupted in 2002 and were not restarted in our pine forest and were restarted in 2016 and are currently conducted in our Gmelin larch forest. Therefore, we based on the three-year consequent measurements for both sites, in close but different observation periods though. Still, we are aware of only three eddy towers in continental Siberia that acted in 2013-2020. Recently, the eddy tower network is being reconstructed. We reviewed accessible papers on fluxes in Siberian forests.

We believe our study is important one that summarizes many ecosystem functions of two contrasting forests therefore we provided many details. We shortened the manuscript by 15-20% and moved some text and figs to Supplementary (marked in yellow) but extended Introduction, Discussion and some literature as recommended. Note, we did not limit pages of our paper in the beginning because the Journal does not require any limits.

Added in Introduction

Among the most recent publications for Siberia are the following: Olchev et al (2022) [17] which studied only energy fluxes (net radiation, latent heat and sensible heat); Masyagina et al (2020) [18] studied soil respiration; and Urban et al. (2019; 2018) [19, 20] studied transpiration and water use efficiency in the same Gmelin larch and Scots pine forests. However, none of them directly and systematically compared the functional permafrost vs. non-permafrost parameters for these two contrasting forest habitats. We compared them and found that these ecosystem parameters are very similar after a decade or so. From the summaries of  the global boreal forests of Virkkala et al. (2021, 2022) [21, 22], we are aware of three forest eddy tower sites across Siberia: one non-permafrost site Zotino (in our study), two permafrost sites Tura (in our study) and Yakutia. A permafrost larch forest in Yakutia (site Spasskaya Pad) was discussed in multiple publications of Machimura et al.[11], Dolman et al.[10], and Maximov et al. [12](see References). None of those publications compared permafrost forest ecosystems to non-permafrost ecosystems in detail. In our present study, we suggested the analyses of multiple ecosystem functional parameters (heat and water vapor fluxes (LE, H, β ); CO2 exchange (NEE, Reco, and GPP); and water use efficiency (WUE)  of two principal forest ecosystems: Scots pine and Gmelin larch forests in interior Siberia.

Recently, Arctic-boreal landscapes, including both permafrost and non-permafrost ecosystems, have been intensively studied based on eddy covariance measurements collected from many eddy sites across the circumboreal zone (Ueyama et al, 2013[23]; Baldochhi et al. 2018[24]; Natali et al. 2019 a, b[25-26]; Virkkala et al, 2022[22]; Watts et al, 2023[27]). As predicted from general circulation models, profound warming is expected at high latitudes; thus Arctic-boreal ecosystems become especially vulnerable to warming, which may transform the Arctic-boreal ecosystems from carbon sink to source (Watts 2023)[27].  Abbot et al. (2016) [28] analyzed 98 permafrost-region experts' evaluations of biomass, wildfire, and hydrologic carbon flux responses to climate change. Experts assessed that the permafrost region will become a carbon source to the atmosphere by 2100.

From  1990 to 2015,  Virkkala et al. (2021) [21]  compiled eddy covariance and chamber measurements of annual and growing season CO2 fluxes of gross primary productivity (GPP), ecosystem respiration (ER), and net ecosystem exchange (NEE) from 148 terrestrial high-latitude sites to analyze the spatial patterns and drivers of CO2 fluxes and test the accuracy and uncertainty of different statistical models. Virkkala et al. (2021, 2022) [21, 22]  developed a standardized monthly database of Arctic–boreal CO2 fluxes (ABCflux) that aggregates in situ measurements of terrestrial net ecosystem CO2 exchange and its derived partitioned component fluxes: gross primary productivity and ecosystem respiration. ABCflux can be used in a wide array of empirical, remote sensing and modeling studies to improve understanding of the regional and temporal variability in CO2 fluxes and to better estimate the terrestrial ABZ CO2 budget. ABCflux is openly and freely available online (Virkkala et al., 2021b)[21] .

Here, we hypothesized that permafrost is a powerful driver that changes site climatology, vegetation cover,  and energy and mass fluxes between vegetation and the atmosphere and thus may change the contemporary C-sink of the boreal forest into C-source if methane emissions exceed C-sink with permafrost retreat, as many studies suggest (Abbott [28]).

  1. The introduction of the article is relatively short, and does not fully describe the research status at home and abroad, and the cited literature is not the latest. It is suggested to comprehensively elaborate the core content of the manuscript, summarize the innovation of this article and solve the scientific problems;

Accepted. We added some recent relevant literature (10 references) that provides additional knowledge for the boreal forest globally and for permafrost ecosystems in particular. Yet, global research was presented in Discussion. The main novel achievements (direct comparisons of ecosystem fluxes between permafrost and non-permafrost forests) were provided in Discussion and Conclusions. We paid much attention to permafrost in the first place and provided much information through the paper on both permafrost itself and consequences that it may cause changing the environment followed by change in the biosphere.

  1. The experimental design was inadequate, and the differences between different places, including water, soil, tree growth and other information, were not clarified; The sampling time of the instrument and the method of data processing were not clearly stated;

We specifically designed out study to compare contrasting sites by environments: climates, soils (permafrost), forests structure. All the details about site environments were given in Introduction and Methods. As we emphasize in Introduction, only one tree species Larix dahurica (later called L. gmelinii) can withstand permafrost. Thus, there is no way to study permafrost impacts on forests of various forest compositions: other tree species just do not grow on shallow permafrost. In our earlier study (Tchebakova et al., 2009), we found that only if permafrost thaws 1.5-2 m deep in summer other tree species may inhabit those sites, e.g. Scots pine in Yakutia, in broad river valley on warm sandy soils.  We gave these details in the text which we consider to be important for the Reader to enlarge their knowledge on permafrost forest ecology.

Regarding various sampling time for our two sites, we based on available data: 1998-2000 in the pine forest and 2004-2008 in the larch forest. After 2002 the pine tower stopped functioning and was never restarted. The larch tower was restarted in 2013. Thus, these periods never overlapped. However, we found for the larch forest that our fluxes estimates of 2004-2008 fell in the same range of those of 2013-2015 (Olchev et al. 2022) which makes our results comparable.

Regarding the method of data processing, please indicate what exactly should we clarify. Because we based on our previous publications we move the Method details to Suppl. in order to shorten the MS as recommended.

  1. It is suggested to move all parts 3.1 and 3.2 of the manuscript to the method part. This is the situation of meteorological factors in the study area, and it is not the result of the study; Move to the method section, and delete a lot, just keep the results;

3.1. was moved to Suppl. as recommended.

About 3.2 we do not quite agree. We may move General climatology to Methods. However, this was our analysis of general climatology based on published data (Reference books…). The site climatology analysis was based on our eddy covariance data and was fully results of our study. Thus we left this portion in the main body of the text.

  1. The discussion on the manuscript is lack of pertinence, and the literature cited for comparative analysis is too old. It is recommended to conduct a systematic discussion on the 3.4, 3.5 and 3.6 parts of the manuscript; This part needs to be added, and the existing content is insufficient;

Accepted. We added 10 newly published papers. However, we cited all the important papers despite the publication year, because we had been learning microclimatology (eddy covariance) science from them and consider them to be important.

Regarding “the systematic discussion on the 3.4, 3.5 and 3.6 parts of the manuscript” we believe we provided more details in both Results and Discussion for 3.4, 3.5 and 3.6 subdivisions. Please advise what  kind discussion should be added

  1. The conclusion of the manuscript is too long, just give the regular result directly; Proper compression is recommended;

Accepted. We left in Conclusions only direct results, however moved the full Conclusion in Suppl. for those who want know a bit more.

  1. The document citation format of the manuscript does not conform to the requirements of the journal.

Of course, the References will be reformatted according to the Journal requirements. We addressed the Editor to allow us to leave author names in references for the first review because we think it is more convenient for reading. If and when the MS is accepted the references would be immediately reformatted.
Thank you again for your thoughtful reading and comments. Hope, we met your recommendations.

On behalf of my co-authors,

Nadezhda Tchebakova

Reviewer 2 Report

This paper focused on the differences in climatology, energy and mass exchange in two types of forests on contrasting habitats in central Siberia based on eddy covariance measurements. Comparative analyses of NEE, NEP and WUE of two forests between the two different habitats. However, I would like to have some questions and suggests:

The title is too long, it is much better to make it into briefed one such as it is unnecessary to add the method for “from eddy …”

In Abstract, Line 15, please add “,” before “respectively”, add a unit after “5.8” in line 26. Line 19, “and”--“with”?

Line 63, please add “to” after “were”. Line 69, why not quantitatively?

Line 72, what is here meaning for “retreat”?

Could you provide a hypothesis for your study in the part of Introduction?

In the part of “Site Description”, it is much better to introduce the larch forest first, and then the pine forest in order to keep the same order for their description in other parts in the text including the abstract.

Line 170-171, I am wondering why you select different periods for the different types of forest? In other words, it may be much better to compare the same or similar stages of years or months.

Line 186, please keep the same level line for your data between the two stations in Table 1.

In Figure 4 and 5, I am wondering why it is quite different years in comparing the related parameters of each tree species? Is there any comparability?

In Figure 5 and 6, “V, VI, VII…” stand for? How many samples or data with “n” ?

In Table 2 and 3, I am wondering why you select the different years for the different tree species. It is much better to compare the same years. Please keep the same level line for your data.

Line 535, the conclusions are too long, they should be concise. In addition, I feel that it is unnecessary to reintroduce the difference of the two site habitats.

Author Response

Rev. 2

Comments and Suggestions for Authors

This paper focused on the differences in climatology, energy and mass exchange in two types of forests on contrasting habitats in central Siberia based on eddy covariance measurements. Comparative analyses of NEE, NEP and WUE of two forests between the two different habitats. However, I would like to have some questions and suggests:

Dear Reviewer,

Thank you for your comments and suggestions which will definitely improve our manuscript.

The title is too long, it is much better to make it into briefed one such as it is unnecessary to add the method for “from eddy …”

Accepted. Starting from “from eddy…” was deleted

In Abstract, Line 15, please add “,” before “respectively”, add a unit after “5.8” in line 26.

Done.

Line 19, “and”--“with”?

Left “and a longer growing season” because “higher values in summer months” do not necessarily extend a growing season

Line 63, please add “to” after “were”. Line 69, why not quantitatively?

 “qualitatively” is a correct word because currently we cannot give numbers how all fluxes will change if permafrost retreats and a larch forest would be replaced by a pine forest. Now, we can only say that C-sink would be greater when and where pine forests would be replaced by permafrost larch forests. But simultaneously methane kept in permafrost would be released so that would be a C-source.  The balance between C-sink and C-source would require some additional research.

Line 72, what is here meaning for “retreat”?

‘Retreat’ is a geographic term for permafrost/glacier moving backward

Could you provide a hypothesis for your study in the part of Introduction?

Our goals followed our hypothesis which was formulated like following and added to Introduction:

Here, we hypothesized that permafrost is a powerful driver that changes site climatology, vegetation cover,  energy and mass fluxes between vegetation and the atmosphere and thus may change contemporary C-sink of the boreal forest into C-source if methane emissions would be greater than C-sink with permafrost retreat as many studies suggest (Abbott [28]).

Our goals were to: 1) compare inter-annual and seasonal variations of energy and mass (water vapor and CO2) fluxes evaluated from eddy-covariance measurements and associated climate variables and to infer the differences in the CO2-exchange driven by various environmental factors in two central Siberian forests located in contrasting habitats: a Larix gmelinii forest on permafrost representative of the vast permafrost zone over tablelands in central Siberia and a Pinus sylvestris forest on comparatively warm sandy soils representative of adjacent West Siberia Plain.  2) try expertly assess a central Siberian boreal forest C-sink change and potential feedbacks to the climate system in “hot spots” of the replacement of Gmelin larch forests on permafrost by the Scots pine forest advance followed the permafrost retreat in a changing climate which would require additional research in interior Siberia

In the part of “Site Description”, it is much better to introduce the larch forest first, and then the pine forest in order to keep the same order for their description in other parts in the text including the abstract.

Fine. Done. Thank you for your keen eye.

Line 170-171, I am wondering why you select different periods for the different types of forest? In other words, it may be much better to compare the same or similar stages of years or months

In Figure 4 and 5, I am wondering why it is quite different years in comparing the related parameters of each tree species? Is there any comparability?

Regarding various sampling time periods for our two sites, we based on available data: 1998-2000 in the pine forest and 2004-2008 in the larch forest. After 2002 the pine tower stopped functioning and was never restarted. The larch tower was restarted in 2013. Thus, these periods never overlapped.

Added.

“Despite sampling time periods were different for our forest sites, we found that our fluxes estimates of 2004-2008 fell in the same range of those of 2013-2015 (Olchev et al. 2022) which makes our results comparable.”

Line 186, please keep the same level line for your data between the two stations in Table 1.

General climatology in Table 1 was moved to Supplementary as recommended by Rev. 1.

Table 2 was corrected.

In Figure 5 and 6, “V, VI, VII…” stand for? How many samples or data with “n” ?

V, VI, VII are months May, June etc. Explanations are given in figure captions.

“N” is a number of days so that for five months this numbers is 153 multiplied by three years 460 days.

Line 535, the conclusions are too long, they should be concise. In addition, I feel that it is unnecessary to reintroduce the difference of the two site habitats.

Accepted. We left in Conclusions only direct results, however moved the extended Conclusion in Suppl. for those who want know a bit more.

Thank you again for your thoughtful reading and comments. Hope, we met your recommendations.

On behalf of my co-authors,

Nadezhda Tchebakova

Reviewer 3 Report

The paper report valuable results on carbon flux in the areas of almost forest limit in Siberia, contrasting permafrost and non-permafrost areas. It also includes comprehensive monitoring on solar energy, vapor water and interesting discussions on carbon flux and water use efficiency. Only several minor points are recommended to improve the manuscript as below.

L 42: ‘deep enough’ for what? Even the authors may mention the permafrost depth directly.

L. 48: Is it aboveground biomass, or include underground?

L. 87: ‘per ha’ or ‘ha-1’. Also, is ‘biomass’ including underground?

L. 103: Is ‘biomass’ including underground?

Table 1 & 2: The rows should be arranged into same levels.

Figure 8 & 9: I recommend some explanations on the curves fitted in the figures.

Author Response

Rev. 3

Comments and Suggestions for Authors

The paper reports valuable results on carbon flux in the areas of almost forest limit in Siberia, contrasting permafrost and non-permafrost areas. It also includes comprehensive monitoring on solar energy, vapor water and interesting discussions on carbon flux and water use efficiency. Only several minor points are recommended to improve the manuscript as below.

Dear Reviewer,

Dear Reviewer,

Thank you for your comments and suggestions which will definitely improve our manuscript.

 L 42: ‘deep enough’ for what? Even the authors may mention the permafrost depth directly.

Added. 1.5-2 m (Tchebakova et al., 2009)

  1. 48: Is it aboveground biomass, or include underground?

Added. “14462 Tg C in their total phytomass, of which 25% is below ground (Shvidenko et al. 2003)

  1. 87: ‘per ha’ or ‘ha-1’. Also, is ‘biomass’ including underground?

Left ‘per ha’, added ‘phytomass’ usually meaning aboveground biomass

  1. 103: Is ‘biomass’ including underground?

Aboveground biomass.

Table 1 & 2: The rows should be arranged into same levels.

Done. Rearranged.

Figure 8 & 9: I recommend some explanations on the curves fitted in the figures.

Our explanations are given for: Fig. 8 in lines 439-447   and Fig. 9 in lines 452-467

Thank you again for your thoughtful reading and comments. Hope, we met your recommendations.

On behalf of my co-authors,

Nadezhda Tchebakova

Round 2

Reviewer 1 Report

The author has overhauled the manuscript, but there are still some problems, including the format of references, and the results should be compressed.

Author Response

Dear Reviewer;

Thank you again for your careful reading and advice.

According to your recommendations we: 1. formatted the References; 2. fixed misprints we found; 3. a journal English editor will fix spell check.

Regarding your recommendation to compress the Results, the authors believe this section is the principal one for which we worked. Therefore, we would like to leave this section as is. We moved 1/3 to Supplementary and 4 pages left. The Reader always may skip some text not interesting for him/her. As a climatologist and geographer I am always interested to know more about the nature, environment etc. of study sites. Therefore I gave enough details in our paper.

Thank you again.

On behalf of the authors, Nadezhda Tchebakova

Reviewer 2 Report

The authors made a wide revision and detailed respond to my previous comments and questions. So I would like to recommend for publication.

Author Response

Dear Reviewer;

Thank you very much for your positive evaluation of our  maanuscript and recommendation  for publication.

On behalf of the authors, Nadezhda Tchebakova